# Establishment and Application of an Indirect Enzyme-Linked Immunosorbent Assay for Measuring GPI-Anchored Protein 52 (P52) Antibodies in *Babesia gibsoni*-Infected Dogs

**DOI:** 10.3390/ani12091197

**Published:** 2022-05-06

**Authors:** Qin Liu, Xueyan Zhan, Dongfang Li, Junlong Zhao, Haiyong Wei, Heba Alzan, Lan He

**Affiliations:** 1State Key Laboratory of Agricultural Microbiology, College of Veterinary Medicine, Huazhong Agricultural University, Wuhan 430070, China; lq1319529141@163.com (Q.L.); xueyan09172022@163.com (X.Z.); dongfang0216@webmail.hzau.edu.cn (D.L.); zhaojunlong@mail.hzau.edu.cn (J.Z.); 2Key Laboratory of Preventive Veterinary Medicine in Hubei Province, Wuhan 430070, China; 3Key Laboratory of Development of Veterinary Diagnostic Products, Ministry of Agriculture of the People’s Republic of China, Wuhan 430070, China; 4Liuzhou Animal Husbandry Station in Guangxi Province, Liuzhou 545025, China; lzsxmz@126.com; 5Department of Veterinary Microbiology and Pathology, College of Veterinary Medicine, Washington State University, Pullman, WA 99163, USA; heba.alzan@wsu.edu; 6Parasitology and Animal Diseases Department, National Research Center, Dokki, Giza 12622, Egypt; 7Tick and Tick-Borne Disease Research Unit, National Research Center, Dokki, Giza 12622, Egypt

**Keywords:** babesiosis, *Babesia gibsoni*, GPI anchor protein, ELISA, diagnosis method

## Abstract

**Simple Summary:**

In this study, a novel BgGPI52-WH antigen was identified and evaluated as an immunodiagnostic candidate. An indirect ELISA was established based on the BgGPI52-WH antigen. The results showed that the iELISA had good sensitivity, specificity and reproducibility, and the signal could be detected as early as the sixth day after infection. Clinical samples were tested using the established method, and 11.41% of the samples were positive. The results suggested that BgGPI52-WH is a good immunodiagnostic marker and the iELISA is a practical method for early diagnosis.

**Abstract:**

*Babesia gibsoni* is a malaria-like protozoan that parasitizes the red blood cells of canids to cause babesiosis. Due to its high expression and essential function in the survival of parasites, the Glycosylphosphatidylinositol (GPI) anchor protein family is considered an excellent immunodiagnostic marker. Herein, we identified a novel GPI-anchored protein named as BgGPI52-WH with a size of 52 kDa; the recombinant BgGPI52-WH with high antigenicity and immunogenicity was used as a diagnostic antigen to establish a new iELISA method. The iELISA had a sensitivity of 1:400, and no cross-reaction with other apicomplexan parasites occurred. We further demonstrated that the degree of variation was less than 10% using the same samples from the same or different batches of an enzyme-labeled strip. It was found that the method was able to detect early infection (6 days after infection) in the sera of the *B. gibsoni*-infected experimental dogs in which antibody response to rBgGPI52-WH was evaluated. Clinical sera from pet hospitals were further tested, and the average positive rate was about 11.41% (17/149). The results indicate that BgGPI52-WH is a reliable diagnostic antigen, and the new iELISA could be used as a practical method for the early diagnosis of *B. gibsoni*.

## 1. Introduction

Babesiosis in dogs is a febrile tick-born disease caused by infections from the *Babesia* species. These parasites are parasitic in the red blood cells (RBCs) of dogs [1,2]. *Babesia gibsoni* (*B. gibsoni*) is one of the main *Babesia* species that causes babesiosis in dogs. Dogs infected with *Babesia* may have hyperacute infections, acute infections, or more commonly chronic infections [3,4]. The main clinical symptoms are fever, hemolytic anemia, and hemoglobinuria. Severe cases can lead to the death of dogs, which brings great spiritual and economic losses to pet owners [5,6]. Babesia, which infects dogs, was first discovered in India in 1910 and has since been reported in countries around the world [7,8]. In China, *B. gibsoni* was reported for the first time in 1987, and the disease was subsequently found throughout the country [9]. In 2017, the strain of *B. gibsoni* in Wuhan (*B. gibsoni*-Wuhan) was reported [10].

In recent years, the incidence of babesiosis in dogs has been increasing, and the number of working and pet dogs has also risen rapidly [10,11]. Therefore, the development of effective diagnosis and treatment of babesiosis is particularly important. The diagnosis of babesiosis in dogs is currently mainly performed by microscopy (microscopic examination “ME”), which cannot be used for the detection of chronic infection or low-infected samples, and is not appropriate for epidemiological studies of large numbers of samples. In addition, external factors such as environment and personnel can interfere with ME as a diagnostic method. It is worth mentioning that dogs infected with *B. gibsoni* have the characteristics of lifelong premunition immunity and easy relapse. Therefore, it is necessary to develop an accurate and simple diagnostic method that can be used in clinical detection.

The first step in establishing a new diagnostic method is to screen excellent candidates as diagnostic antigens [12]. Studies have shown that, when parasites invade the host, parasite surface proteins are most easily discovered by the host immune system, and thus play a major role in the host’s immune response [13]. So, parasite surface proteins are generally recognized as important candidates for diagnostic antigens [14,15,16]. GPI is a glycolipid structure added to the carboxyl terminus of most eukaryotic proteins after translation [17,18]. This modification is responsible for anchoring the protein on the outer side of the cell membrane to become the surface protein of the parasite [19,20,21,22]. Moreover, the GPI site of the GPI-anchored protein is released from the surface of the cell membrane after being cut and exposed to the host immune system directly, which theoretically can effectively cause the immune protection mechanism of the host [22]. GPI-anchored proteins are reported to be abundant on the membranes of apicomplexan parasites. They have been shown to have good immunogenicity and are suggested to be important virulence components of parasites [20,23]. For example, the merozoite surface proteins (MSPs) in *Plasmodium falciparum* (*P. falciparum*), the surface antigens (SAGs) in *Toxoplasma gondii,* and the merozoite surface antigens (MSAs) in *Babesia* have typical GPI anchor modification structures [24,25,26]. Therefore, this type of GPI-anchoring protein is considered to be an excellent candidate vaccine and detection antigen for apicomplexan protozoa [17,19].

Previous studies have shown that GPI-anchored proteins have a great immunogenicity in many *Babesia* spp. For example, the merozoite surface antigen (MSA-2C protein), as a highly immunogenic protein, has been successfully used in the development of serological diagnostic methods such as iELISA of *B. bovis* [27,28,29]. For *B. microti*, it was found that the BmGPI12 protein alone or in combination with other *B. microti* GPI-anchored proteins has great potential as a diagnostic biomarker, in addition to being a drug target [19]. Therefore, a class of proteins containing GPI anchor sites may play an important role in the detection of Babesiosis. There is no complete diagnostic kit for *B. gibsoni* in China.

In this study, the BgGPI52-WH-anchored protein was screened using bioinformatics methods, then cloned and expressed to identify its antigenicity and immunogenicity as a potential diagnostic antigen. Finally, an iELISA diagnostic method was established using the BgGPI52-WH-anchored protein to explore its possibility as an immunodiagnostic marker. This provides a theoretical basis for the clinical detection, treatment and prevention of *B. gibsoni*, and provides a candidate target for the diagnosis of *B. gibsoni*.

## 2. Materials and Methods

### 2.1. Parasites and Animal Experiments

*B. gibsoni*-Wuhan strain was stored in liquid nitrogen in our laboratory, the State Key Laboratory of Agricultural Microbiology, Huazhong Agricultural University, China. We purchased three healthy beagles of about one year of age from the Anlu Laboratory Animal Center. Then, 5 mL of *B. gibsoni*-infected blood (~2.5 × 10^8^ parasites) was injected into the experimental beagles by subcutaneous injection. PCR and microscopy were used to detect the parasite infection. In the early stage of infection, the specific fragments of *B. gibsoni* were amplified by conventional PCR to test whether the experimental dogs were successfully infected. When the positive bands of *B. gibsoni* were detected by PCR, they were stained with Giemsa solution and directly observed for *B. gibsoni* under a microscope.

### 2.2. Bioinformatics Analysis

The GPI-anchored proteins of the genus *Piroplasma* (such as *B. microti*, *B. bovis*, etc.) were researched online using the NCBI, Uniprot and PirolplasmaDB databases to identify possible surface proteins of *B. gibsoni* [30]. The signal peptides and transmembrane regions of the surface proteins of *B. gibsoni* were predicted using bioinformatics methods. The GPI anchor points of the candidate proteins were predicted by bioinformatics software such as GPI-SOM, PredGPI and bigPI to prove that these candidate proteins were GPI-anchored proteins. The acquired results were compared with the genome database of the *B. gibsoni*-Wuhan strain (data unpublished), and the GPI-anchoring protein gene of the *B. gibsoni*-Wuhan strain was obtained. Bioinformatics methods were subsequently performed to analyze the presence of the signal peptide, transmembrane region, GPI anchor positions, and B cell epitopes. The bioinformatics analysis website and the software used in this research are presented in Table 1.

### 2.3. Extraction of Total RNA and gDNA

The total RNA was extracted from the infected RBCs (iRBCs) using TRIzol^®^RNA (Thermo Fisher, Waltham, CA, USA). Then, we used the PrimeScript™ RT reagent kit with the gDNA Eraser kit (TaKaRa, China) to reverse transcribe the extracted RNA to obtain *B. gibsoni* cDNA, which we then stored at −80 °C. 

Genomic DNA (gDNA) was extracted from the iRBCs of dogs using the QIAamp DNA Kit (Qiagen, Shanghai, China). Follow-up experiments were performed following the success of the quality test. After concentration measurements, the DNA was stored at −20 °C.

### 2.4. Cloning, Expression and Purification of rBgGPI52-WH

The BgGPI52-WH gene sequence was obtained by PCR with *B. gibsoni* gDNA and cDNA as templates, using the specific primers of the BgGPI52-WH gene (Table 2). The PCR product was detected by 1% agarose gel (TSINGKE Biological Technology, Beijing, China), and subsequently recovered by the Easy Pure^®^ PCR Purification Kit (TransGEN, Beijing, China). The recovered products were connected and transformed into the pEASY-Blunt vector (TaKaRa Biotechnology, Beijing, China), and the sequence of BgGPI52-WH gene was confirmed after sequencing. After correct sequencing, *BamHI* and *XhoI* were used as restriction sites, and the pET-28a expression vector was used for protein expression. The recombinant protein with His-tag (rBgGPI52-WH) was expressed in *E. coli* BL21 cells. The normal expression of BL21 was induced after extended culture, and the protein was identified by SDS-PAGE gel pattern with His-tag and its expression in the supernatant. The supernatant was centrifuged at 4 °C after pressure crushing. The supernatant was combined with the nickel-affinity chromatography column (GE Healthcare, Uppsala, Sweden) then eluted and purified with imidazole at different concentrations. The eluents of each gradient were recovered and visualized by SDS-PAGE. The concentration was determined BCA method (Beyotime, Shanghai, China) according to manufacturer’s instruction. After protein purification with a nickel column, the target protein was obtained and stored at −80 °C until further use.

### 2.5. Production of Mouse Anti-rBgGPI52-WH Immune Serum

Five KunMing mice were used for the preparation of antibodies against rBgGPI52-WH. The target protein and Freund’s complete adjuvant (Sigma, Shanghai, China) was mixed in a ratio of 1:1 for protein emulsification, and then each mouse was subcutaneously immunized with 100 μg of rBgGPI52-WH. Thereafter, on days 14, 28, 42, 56, 70 and 84, the booster doses were administrated. The serum samples were collected one week after the sixth immunization.

### 2.6. Preparation of B. gibsoni Lysates

Next, 1 mL of fresh red blood cells (RBCs) was collected from the *B. gibsoni*-infected dogs and resuspended in an equal volume of PBS, and 18 mL of RBCs lysis buffer was added (Tris/EDTA/NaCl). Then, the supernatant was preheated for 5 min in a 37 °C water bath and centrifuged at 2450 r/min for 5 min. After collecting the supernatant, the supernatant was centrifuged at 12,000 r/min for 20 min to retain the pellet. The obtained pellet was washed by PBS 3 times. Then, 5 mL PBS was added to resuspend the pellet, then centrifuged at 15,000 r/min for 20 min. We discarded the supernatant then resuspended the pellet with 1 mL of PBS. The lysate of *B. gibsoni* merozoites was used directly, or stored at −20 °C.

### 2.7. Western Blot Analysis

To detect the antigenicity of BgGPI52-WH, the rBgGPI52-WH protein was electrophoresed onto SDS-PAGE gel. The primary antibodies were the positive serum of *B. gibsoni*-infected dogs and the negative serum of pre-infection dogs. The secondary antibody was goat anti-canine IgG (HRP) (Southern Biotech, Birmingham, USA). To preliminarily determine the immunogenicity of rBgGPI52-WH, SDS-PAGE gel electrophoresis was performed with erythrocyte lysates from normal dogs and *B. gibsoni*-infected dogs. Mouse anti-BgGPI52-WH polyclonal antibodies (PcAb) and pre-immune mouse serum were used as the primary antibody, in addition to the secondary antibody of goat anti-mouse IgG (HRP) (Southern Biotech, USA). The electrochemiluminescence (ECL) method was used for developing the target band.

### 2.8. Establishing iELISA Based on rBgGPI52-WH

In this study, the rBgGPI52-WH target protein was coated on the microplate as previously described [29]. The primary antibody comprised the serum of healthy dogs and *B. gibsoni*-infected dogs, and the secondary antibody was goat anti-canine IgG (HRP) (Southern Biotech, USA). The optimization steps were as follows: (1) the optimum concentration of antigen envelope and serum dilution ratio were screened. The purified rBgGPI-WH protein was diluted to different concentrations with the coating buffer according to doubling dilution, and the serum was also diluted to different concentrations by doubling dilution, and then screened by cross-square titration. (2) The second step was to screen the best time for antigen blocking. The protein was coated with the optimal coating concentration screened above, and the protein was blocked at different times while fixing other experimental conditions. The closing time at the maximum P/N value (P/N value = (mean value of positive OD_630_/mean value of negative control OD_630_)) was regarded as the best antigen blocking time. (3) Next, we screened the optimal incubation time for the primary antibody. Screening was performed by adjusting the serum action time of the different gradients after determining the optimal coating concentration of the antigen, the optimal serum dilution ratio, and the optimal antigen blocking time. (4) Likewise, under the fixed conditions of the other experimental conditions, single factors were screened one by one. The screening work consists of the best working concentration of the secondary antibody, the best incubation time, and the best working time of the color substrate solution. After multiple optimizations, the established iELISA method was used to detect the infection using multiple sera, and the cut-off value was determined after a fixed formula calculation.

### 2.9. Sensitivity and Speficity of the iELISA Assy

The sensitivity of the established iELISA method was determined by using *B. gibsoni*-positive sera. We diluted the known positive sera and negative sera from our laboratory in the ratio of 1:200, 1:400, 1:800, 1:1600, 1:3200, and 1:6400 with the incubation solution to perform the iELISA test. After terminating the reaction, we read it with a microplate reader. Values of S/P (S/P value = mean value of sample OD_630_/mean value of positive control OD_630_) were calculated to determine the sensitivity.

For the specificity assay, seven different sera, including *B. gibsoni*, Toxoplasma gondii (*T. gondii*), Echinococcus granulosus, Strongyloides stercoralis (*S. stercoralis*), Rabies virus (RABV), Canine Parvovirus (CPV), and *Babesia canis* (*B. canine*) were used. Serum from healthy dogs without any known disease was considered as a negative serum. The iELISA test was performed according to the optimized conditions. After the reaction was terminated, the plates were read at the OD_630_ nm value with a microplate reader. The S/P value were calculated, and the negative or positive values were estimated according to the cut-off value standard to the established test in this study to determine the specificity of the ELISA method.

### 2.10. Repeatability of the iELISA Assay

Four positive sera and two negative sera known in our laboratory were used to perform the intra-batch reproducibility experiments and the inter-batch reproducibility experiments. After the reaction was terminated, reading was established at the OD_630_ nm value with a microplate reader. The average (x) and standard deviation (SD) of each serum were calculated at OD_630_ nm, and the coefficient of variation was calculated using the following formula CV = (SD/x) × 100%.

### 2.11. Detection of Antibodies against BgGPI52-WH in Experimental Infected Dog Samples

Three experimental beagles (A, B, C) were infected with the *B. gibsoni*-Wuhan strain under laboratory conditions. Serum samples from the infected dogs were collected at intervals of 3 or 7 days, up to 101 days after infection. The antibodies in the serum of the *B. gibsoni*-infected beagles were detected by the established iELISA to study the production of anti-BgGPI52-WH antibodies in beagles infected with *B. gibsoni*.

### 2.12. Evaluating the iELISA by Clinical Sample Testing

A total of 149 clinical sera samples with clinical symptoms of fever and anemia were donated by veterinary hospital in Wuhan, China. The clinical sera samples were tested by iELISA based on rBgGPI52-WH and rBgSA1-WH (unpublished data), respectively, and the iELISA detection results of the two proteins were compared. All the performed work was repeated 3 times. Data analyses were performed using Excel 2019 (Microsoft Corporation, Redmond, WA, USA) and graph generation was performed using GraphPad Prism 8 (Graph Pad Software, San Diego, CA, USA). S/P values were calculated by Excel 2019 (Microsoft Corporation, Redmond, WA, USA), and the figures were generated by GraphPad Prism 8 (Graph Pad Software, San Diego, CA, USA) by using XY tables (points only graph).

## 3. Results

### 3.1. Gene Sequence Analysis of BgGPI52-WH

Five GPI anchor proteins of *B. gibsoni* were screened by bioinformatics. The results of the analysis showed that these five candidate antigens were potential diagnostic candidates. Previous research on one of the GPI-anchored proteins, BgGPI47-WH, has shown that it is a good diagnostic antigen, and related research content has been published [29]. In the present work, the gene sequence of BgGPI52-WH was obtained by PCR amplification from *B. gibsoni* gDNA and cDNA with specific primers (Table 1). The full length of the BgGPI52-WH gene is 1453 bp, and it contains an intron of 37 bp (Figure 1a). A signal peptide was predicted in the first 1–19 aa of the BgGPI52-WH amino acid sequence, containing at least two transmembrane regions (3–21 aa, 454–471 aa) at the C-terminal. The GPI anchor point was located at the C-terminus, and the length was approximately 21–22 aa (Figure 1b). The epitope of BgGPI52-WH was predicted by DNAStar software. The results revealed that the BgGPI52-WH protein has a good epitope, and could be used as a high-quality antigen molecule for the development of novel diagnostic methods (Figure 1c).

### 3.2. Cloning and Expression of BgGPI52-WH Recombinant Protein

The full ORF of BgGPI52-WH gene contained 1416 bp, encoding 471 aa with a molecular weight of ~52 kDa. The recombinant protein was mainly expressed in the supernatant of *E. coli* BL21 (DE3) lysate induced by isopropyl beta-D-thiogalactopyranoside (IPTG) (Figure 2). A specific band was observed around 52 kDa by a 12% SDS-PAGE analysis, which is consistent with the predicted size (Figure 2). For protein extraction, the rBgGPI52-WH was purified using a Ni-agarose column and stored in −80 °C for the subsequent experiment.

### 3.3. Western Blot Analysis of rBgGPI52-WH Protein

For the evaluation of the potential antigenicity and immunogenicity of BgGPI52-WH, we performed a Western blot analysis for rBgGPI52-WH (Figure 3). The infected *B. gibsoni* serum and the noninfected *B. gibsoni* serum were used as primary antibodies to react with rBgGPI52-WH, respectively. A band of ~52 kDa was observed using the positive serum, and no reaction was detected using the negative serum (Figure 3a). Interestingly, the polyclonal antibody (PcAb) of mouse anti-BgGPI52-WH could recognize the native BgGPI52-WH from the lysate of *B. gibsoni*-infected RBCs, and the size was ~52 kDa. No signal was detected in the *B. gibsoni*-free RBCs lysates (Figure 3b).

### 3.4. Optimization of Experimental Conditions of Indirect ELISA

Based on the data above, we attempted to establish an iELISA method using the newly identified BgGPI52-WH. After the optimization, the best experimental conditions were acquired. The optimal coating concentration of the rBgGPI52-WH was determined as 1 μg/mL, and the optimal blocking time was 30 min using 1% BSA. The best dilution ratio and reaction time for the primary antibodies were determined as 1:200 and 60 min, respectively. The best dilution ratio of secondary antibodies was 1:4000, when the incubation time was 30 min. The reaction time of TMB substrate solution was optimized as 10 min. The cut-off value was calculated using the following equation: X + 3SD. The critical value was determined as 0.296. Hence, the S/P value (>0.296) was used as a reasonable standard to distinguish that of *B. gibsoni*-infected and *B. gibsoni*-free samples. 

### 3.5. Sensitivity and Specificty Detection of BgGPI52-WH iELISA

For evaluating the reliability of the established iELISA, we used the new method to examine five *B. gibsoni*-infected samples to evaluate its sensitivity and specificity detection ability. These samples were further diluted by the doubling dilution method, and the S/P values were calculated after reading with a microplate reader at OD_630_ nm. According to the S/P value, the sensitivity of the novel iELISA was greater than 1:400 (Figure 4). In addition, the established iELISA method was also used to detect the positive sera of *T. gondii*, *Echinococcus granulosus*, *S. stercoralis*, *RABV*, *CPV* and *B. canine*. After the reaction was terminated, the absorbances were acquired at OD_630_ nm and the S/P values were calculated. The results demonstrated that the method based on BgGPI52-WH had high specificity, and no cross reaction was observed with that of the positive sera of *T. gondii*, *Echinococcus granulosus*, *S. stercoralis*, *RABV*, *CPV* or *B. canine*. (Figure 5). Together, the results indicated that BgGPI52-WH is a reliable diagnostic antigen, and the novel iELISA method could be used as a cost-effective way to diagnose *B. gibsoni*.

### 3.6. Repeatability Detection of iELISA

The repeatability of the iELISA was carried out based on the intra-batch and inter-batch repeatability experiments. Four positive sera and two negative sera samples were used in the following experiment, and the repeatability was tested using the strategy of the same batch and different batches of enzyme plates. After the reaction was terminated the absorbances were acquired at OD_630_ nm. The mean value (x) and standard deviation (SD) of the absorbance of each serum were calculated, and the consequent values were further used to determine the coefficient of variation (CV). By comparison, we found that the degree of variation of the same sample from the same or different batch of enzyme-labeled strips was less than 10%, and the results indicated that the new iELISA had good repeatability (Table 3 and Table 4).

### 3.7. iELISA Evaluation Using Clinical and Experimental Infected Samples

For evaluating the application value of the new iELISA, we used the established iELISA method to detect the antibodies from three experimental dogs in different infection stages of *B. gibsoni*. Based on the continuous detection of parasitized rate by *B. gibsoni*, we found that the percentage of parasitized erythrocytes (PPEs) was 0.3% on day 9, 4% on day 10, and up to 33% on day 19. To control the infection, we treated the dogs with intravenous azithromycin, glucose, and other nutrients. Following the treatment, PPE decreased rapidly until it reached almost zero by a microscopic examination, and the dogs entered the lifetime status of *B. gibsoni* infection. As a result, the antibody levels in these dogs will remain high for a long time. In the study, PPEs were combined for a comprehensive analysis, and the new method based on rBgGPI52-WH distinguished the positive sera of three experimental dogs during the early infection stage (Figure 6). Interestingly, the method could detect the *B. gibsoni*-infected sera at day 6 of the early infection stage. The results demonstrated that BgGPI52-WH could detect the positive sera of dogs at the initial stage of infection, and this early detection will contribute to the clinical treatment of babesiosis. In addition, a total of 149 clinical pet dog blood samples were donated by a veterinary hospital in Wuhan, China. These blood samples were tested using the iELISA methods based on BgGPI52-WH and BgSA1-WH, respectively. The results showed that the positive rate of the clinical samples was ~11.41%, and the agreement rate of two iELISA methods was 83.89% (Figure 7).

## 4. Discussion

*Babesia gibsoni* is one of the most widespread sources of *Babesia* infection in dogs. Several different geographic isolates have been obtained from South Asia, East Asia and Southeast Asia [31,32]. It is reported that, with the development of globalization, the infection, morbidity, and mortality rates of babesiosis in dogs in China are on the rise. Among the sera samples of working dogs (26.1%), fighting dogs (39.8%), and pet dogs (3.47%), the highest positive rate was found in the sera samples of the fighting dogs [33]. This could cause economic losses to pet owners and threaten public health. It is therefore necessary to establish a detection method with good specificity and sensitivity for early diagnosis to prevent the outbreak of babesiosis. Microscopy is simple and convenient, and has a high detection rate in acute infections, but chronic infections or carrier dogs cannot be detected. Molecular diagnostic methods have high sensitivity and specificity, but low clinical applicability and high cost, in addition to being time consuming. Compared with other molecular biological detection methods, the ELISA method is easier to operate and more specific for the detection of a large number of samples, especially in large-scale epidemiological investigations.

The key to establishing an effective diagnostic method is to screen high-quality immunodiagnostic markers. The BgTRAP (TRAP of *B. gibsoni*) antigen detected by indirect ELISA has been recognized as the best immunodiagnostic marker for this infection. However, the recombinant expression of BgTRAP is difficult and the sensitivity is not ideal, which requires further research and the development of more prospective antigens [34]. The GPI anchor site helps the antigen anchored to the cellular membrane, and works as a surface protein or secreted protein which plays an important role when the protozoa invade the host. It also has potential to be used as an immunodiagnostic marker [35,36,37]. Therefore, this study was based on the fact that there are no large-scale clinical diagnosis methods for the early stage of Babesiosis in dogs [38]. The GPI-anchored protein BgGPI52-WH was screened by bioinformatics methods and experimental validation. The rBgGPI52-WH protein was expressed and purified. Western blot analysis showed that it has good antigenicity and immunogenicity, and could be used as an immunodiagnostic marker. The above results of the characterization of the BgGPI52-WH protein were similar to BgP50 in the Japanese strain [39]. A method of iELISA detection based on BgGPI52-WH was established. The sensitivity, repeatability, and specificity of the assay were evaluated, as were sera samples from experimental dogs of different infection cycles, and large clinical sera samples.

During natural canine infection with *B. gibsoni*, coinfection with other related parasites usually occurs [40]. It is important to distinguish between pathogens based on epidemiology, and identify other parasites that are closely genetically related to infection by *B. gibsoni*. In this regard, we evaluated the cross-reaction of the diagnostic antigen BgGPI52-WH with the positive sera of other pathogens and found that the iELISA method established with BgGPI52-WH did not react with the positive sera of *Toxoplasma gondii*, Hydatid Canis, Roundworm coelata, rabies virus, canine parvovirus or *Babesia canis*. This test was performed to avoid false positive misdiagnosis results for symptomatic treatment, and the results proved the specificity of BgGPI52-WH for only *B. gibsoni*. The repeatability and sensitivity experiments proved that the method had good repeatability and sensitivity. To detect the serum status of the experimental dogs in different infection cycles, the iELISA method established by BgGPI52-WH, PCR, and microscopy were used to detect the infected dogs. The results showed that the iELISA method established by BGGPI52-WH and PCR were suitable for the early stage of infection. Microscopic examination was not suitable. BgGPI52-WH antibodies could be detected on the sixth day at the earliest, and at the latest on the eighth day of infection. The specific antibody reaction could be detection detected up to day 101, but PCR and microscopy did not work during this period, suggesting that the established method based on BgGPI52-WH is useful for chronic infection. Moreover, PCR and microscopy are more complicated than iELISA, so they are not applicable in general clinics.

In clinical applications, diagnosing the disease in its early stage is important for treatment and disease control [41,42,43]. One hundred and forty-nine clinical samples donated by a veterinary hospital in Wuhan, China, were tested with two different diagnostic antigens. The positive rates of BgGPI52-WH and BgSA1-WH were 11.41% and 17.45%, respectively. The coincidence rate of the two detection methods was 85.23%. This indicates that the test based on the diagnostic antigen BgGPI52-WH had lower false positives, and is thus more consistent with tests suitable for clinical samples.

## 5. Conclusions

In conclusion, an antigen named BgGPI52-WH was identified from the *B. gibsoni*-Wuhan strain. Antigenicity and immunogenicity evaluation demonstrated that it was a potential diagnostic marker. An iELISA with high sensitivity and specificity was established based on the recombinant BgGPI52-WH, which can be used for the clinical diagnosis of early and chronic infections. Early detection means that babesiosis in dogs can be treated quickly which, in turn, reduces the mortality and infection rate of the disease. We believe the iELISA based on the BgGPI52-WH antigen is conducive to the prevention and control of babesiosis in dogs.

## Figures and Tables

**Figure 1 animals-12-01197-f001:**
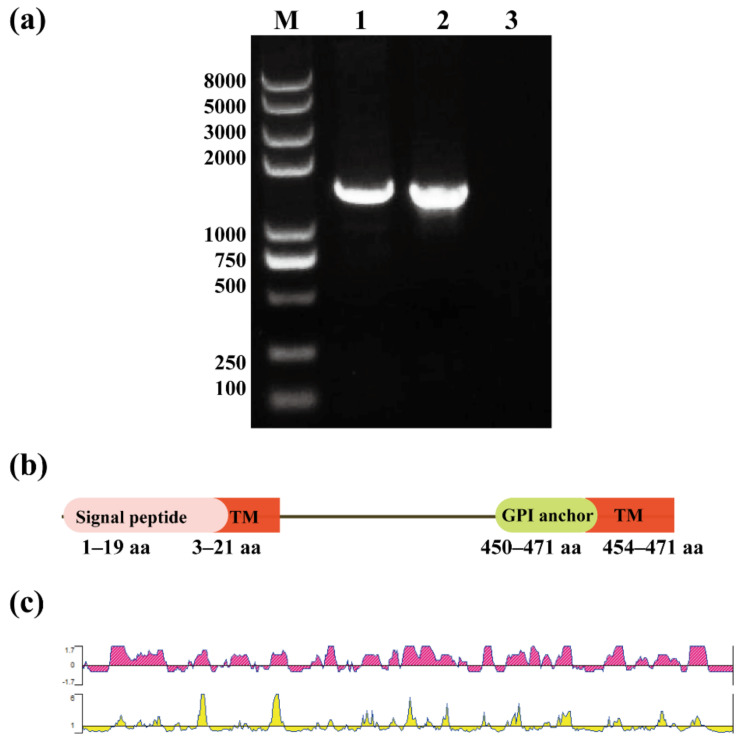
Sequence analysis of BgGPI52-WH. (**a**) Identification of target gene sequences. Lane M—marker; lane 1—the BgGPI52-WH target band amplified from *B. gibsoni* gDNA; lane 2—the BgGPI52-WH target band amplified from *B. gibsoni* cDNA; lane 3—control group. (**b**) A schematic diagram showing the TM domain, signal peptide (SP) and GPI anchor site of the target protein sequence. (**c**) Prediction of the antigen epitopes of the BgGPI52-WH protein. Purple represents the antigenic index; the higher the antigenic index, the better the antigenicity. Yellow represents the probability index of protein on the surface.

**Figure 2 animals-12-01197-f002:**
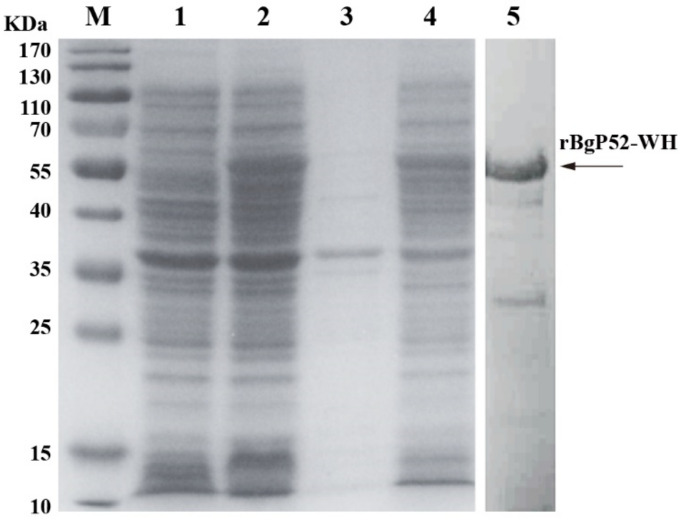
Prokaryotic expression of recombinant protein BgGPI52-WH. M—marker; 1—non-induced BgGPI52-WH; 2—induced BgGPI52-WH; 3—induced BgGPI52-WH precipitate; 4—induced BgGPI52-WH supernatant; 5—purified recombinant BgGPI52-WH protein.

**Figure 3 animals-12-01197-f003:**
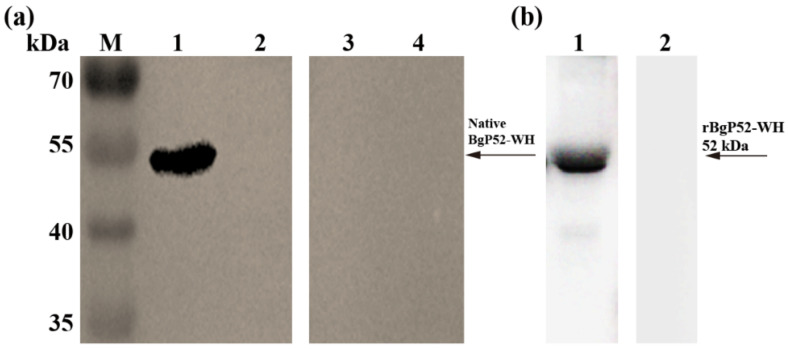
The antigenic properties of BgGPI52-WH were detected by Western blot. (**a**) Immunogenicity detection of *B. gibsoni* BgGPI52-WH protein. Lane M—marker; lane 1, 3—the iRBC lysates of dogs detected by mouse anti-BgGPI52-WH serum; lane 2, 4—the RBC lysates of uninfected dogs detected by mouse anti-BgGPI52-WH serum. (**b**) Antigenicity detection of *B. gibsoni* BgGPI52-WH protein. Lane M—marker; lane 1—recombinant BgGPI52-WH protein detected by the serum of *B. gibsoni*-infected dogs; Lane 2—recombinant BgGPI52-WH protein detected by normal serum from uninfected dogs.

**Figure 4 animals-12-01197-f004:**
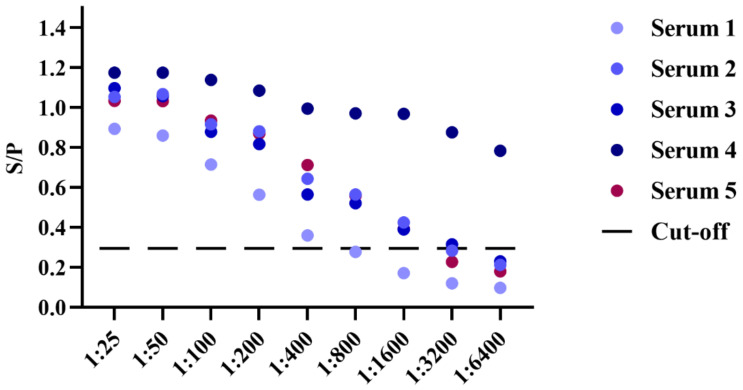
Sensitivity of indirect ELISA to BgGPI52-WH. Serum1–Serum5—five known positive sera; cut-off—cut-off value obtained from the above experiment (0.296).

**Figure 5 animals-12-01197-f005:**
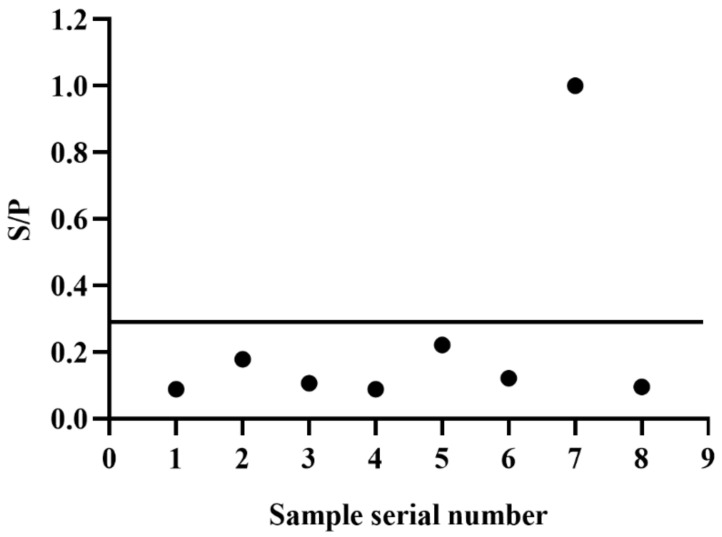
Indirect ELISA specific test. Bands 1–7 reflect serum from dogs infected with *Toxoplasma gondii*, *Echinococcus granulosus*, *Strongyloides stercoralis*, *Rabies virus*, *Canine Parvovirus*, *Babesia canis*, and *Babesia gibsoni* (positive control), respectively. Point 8—negative control.

**Figure 6 animals-12-01197-f006:**
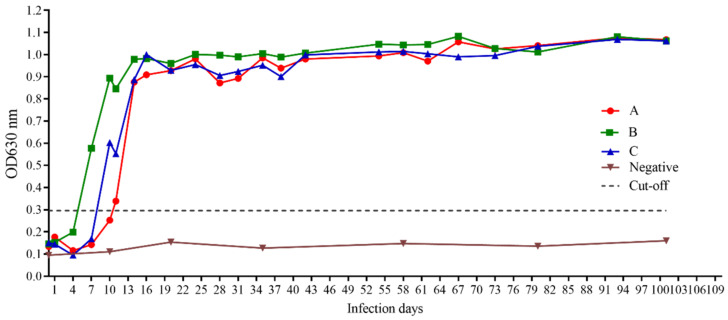
The antibody change curve of BgGPI52-WH detected by BgGPI52-WH-ELISA. The three experimental beagles infected with *B. gibsoni* under laboratory conditions are labelled A, B, and C.

**Figure 7 animals-12-01197-f007:**
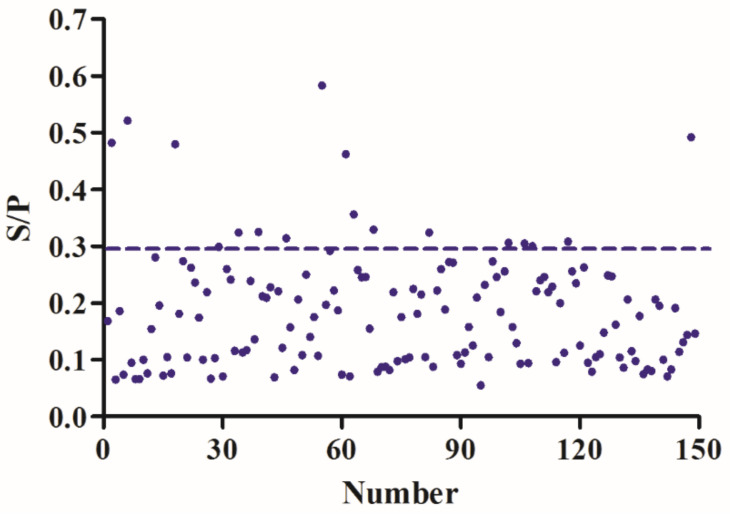
The detection of BgGPI52-WH-ELISA from a clinical sample. The dashed read line indicates the cut-off value. The critical value is X + 3SD = 0.296. Results criteria: serum samples with S/P values greater than 0.296 could be judged as positive. The positive rate after the ELISA test was 11.41% (17/149).

**Table 1 animals-12-01197-t001:** Bioinformatics analysis websites and software.

Function Category	Website/Software
Nucleotide sequence	https://www.ncbi.nlm.nih.gov/, accessed on 12 December 2019
	http://piroplasmadb.org/piro/, accessed on 12 December 2019
	http://www.uniprot.org/, accessed on 12 December 2019
Sequence alignment	http://mafft.cbrc.jp/alignment/server/index.htmL, accessed on 12 December 2019
	http://www.ncbi.nlm.nih.gov/blast/, accessed on 12 December 2019
SignalP	http://www.cbs.dtu.dk/services/SignalP/, accessed on 12 December 2019
Transmembrane prediction	https://embnet.vital-it.ch/software/TMPRED_form.html, accessed on 12 December 2019
ProtScale	https://web.expasy.org/protscale/, accessed on 12 December 2019
GPI anchor site prediction	http://mendel.imp.ac.at/gpi/plant_server.htm, accessed on 12 December 2019
	http://gpi.unibe.ch/, accessed on 12 December 2019
	http://gpcr.biocomp.unibo.it/predgpi/, accessed on12 December 2019
B cell epitope prediction	DNASTAR
Primer design	Clone Manager

**Table 2 animals-12-01197-t002:** Primers used for the amplification of the partial BgGPI52-WH gene.

Primers	Primer Sequences (5′–3′)	Restriction Enzyme
BgGPI52-WH-F	5′-ATGAGACTAGTTCGTGCATTCC-3′	
BgGPI52-WH-R	5′-TTAAAATACAGCGACAGCCACAG-3′	
BgGPI52-WH-*BamH* I-F	5′-CAGGATCCACTGGTGATGGGAATATGACAG-3′	*BamH* I
BgGPI52-WH-*Xho* I-R	5′-TCCTCGAGTTAAAATACAGCGACAGCCACAG-3′	*Xho* I

**Table 3 animals-12-01197-t003:** Intra-batch repeatability test.

	Average Value	SD	CV%
Serum 1	0.943	0.038	4.03
Serum 2	0.989	0.028	2.83
Serum 3	0.501	0.043	8.58
Serum 4	0.088	0.003	3.41
Serum 5	0.121	0.007	5.79

**Table 4 animals-12-01197-t004:** Inter-batch repeatability test.

	Average Value	SD	CV%
Serum 1	0.992	0.043	4.335
Serum 2	0.997	0.025	2.508
Serum 3	0.588	0.013	2.192
Serum 4	0.057	0.004	7.144
Serum 5	0.133	0.010	7.519

## Data Availability

The sequence was submitted to NCBI GenBank (Accession number: MZ773409). The data involved in this study are available through the corresponding author upon reasonable request.

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
