# Peer review of "Establishment and Application of an Indirect Enzyme-Linked Immunosorbent Assay for Measuring GPI-Anchored Protein 52 (P52) Antibodies in Babesia gibsoni-Infected Dogs"

_animals, 2022, doi:10.3390/ani12091197_

Round 1
Reviewer 1 Report
the manuscript entitled “Establishment and application of an indirect enzyme-linked immunosorbent assay for measuring GPI-anchored protein 52 (P52) antibody from Babesia gibsoni infected dog” identified a novel BgGPI52-WH antigena s a immunodiagnostic candidate. The article is well structured, although there are several writing mistakes. A linguistic revision is required.
For example:
line 66 – “So” capital
line 86 – change a ddition with “addition”
line 258 - delete the point after -80 °C
line 284 – change accquired with ”acquired”
line 291 - what words is “reseaonble”?
line 303 – correct “ toghether”
lines 316, 339 and 340 – change “ serums” with “sera”
line 321 – change “determin” with “determinate”
line 328 – change title “Evaluate the iELISA by using experimental infected samples and clinical samples” with “iELISA evaluation by using experimental infected samples and clinical samples”
line 343 – change “clinlic” with “clinic”
line 356 – change “ canies” with “dogs”
lines 380 -383 - change “A method of iELISA detection based on BgGPI52-WH was established. And the sensitivity, repeatability, and specificity of the assay were evaluated, as well as serum samples from experi mental dogs at different infection cycles and large clinical serum samples” With “A method of iELISA detection based on BgGPI52-WH was established: the sensitivity, repeatability, and specificity of the assay were evaluated, as well as serum samples from experi mental dogs at different infection cycles and large clinical serum samples”.
Author Response
Thank you very much for your valuable suggestions on this manuscript. The attachment is my modification and reply. I hope the revised manuscript can meet the requirements for publication.

Reviewer 2 Report
Good presentation of results in a manuscript.
In line# 83, minor change of spelling is required.
In line# 86,405,409 minor grammatical changes are required.
Need to add reference in introduction ( line 62)
Need to add latest references in introduction.
Author Response

(The authors gave the same response as above.)

Reviewer 3 Report
Authors have presented interesting data on an potential diagnostic technique to detect Babesia gibsoni infection in dogs.
General comments:
Spelling mistakes need to be addressed through out the manuscript. E.g. Lines, 100, 118 (table 1), 157, 342
Scientific names should be italicised through out the manuscript. E.g. Lines: 85, 197-199
The manuscript needs an extensive attention to the English grammar and the writing style. E.g. Lines: 99-100, 168, 175-178, 395-396
Abbreviations need to be expanded at the 1st instance that they are used in the text. E.g. P/N and S/P
Specific comments:
Lines 102-103: Brief description of PCR and microscopic methods that were carried out to detect the parasitaemia needs to be included.
Line 110: The information such as the version and the licensed country of used software should be indicated. This also applies to other relevant sections of the manuscript.
Line 113: Tool names of the online applications needed to mentioned wherever applicable.
Line 118, Table 1: Right column doesn't align with the left column. E.g. Two websites listed for the sequence alignment.
Line 136: Protein expression and purification protocols need to be mentioned in detail.
Section 2.12: It is unclear whether the 149 samples were randomly selected or selected based on related clinical signs. However, a parallel available test of highest sensitivity or the gold standard method should be used to validate the results and also to show the superiority of the proposed method to currently available methods.
Lines 233-235: Information on specificity and the sensitivity values need to be included in this statement.
Lines 303-395: This statement should be supported with the diagnostic sensitivity of the early detection in currently used methods in order to justify the proposed method is more sensitive in early detection.
Lines 343-345: The sentence is unclear.
Line 365: Other available molecular biological methods should be mentioned.
Line 375-376: Authors need to provide evidences with published literature to support this statement.
Lines 376: This statement referring to the reference 36 is unclear.
Lines 381-383: This sentence is unclear.
Lines 401-403: This statement has less value as it is based on another unpublished data.
Author Response

(The authors gave the same response as above.)

Reviewer 4 Report
Authoer; Delete “and”
Line 42; Did you need “obligately”?
Line 41-62; In this text, canine babesiosis and B. gibsoni infection are mixed. This time, you are focusing on B. gibsoni, so I would loke you to focus on B. gibsoni from Line 52.
Line 52-62; PCR tests are also common for the diagnosis of canine babesiosis and are extremely sensitive and specific. Is there any advantage to using iELISA over PCR testing?
Line66; “so” →“So, ”
Line 85; B. microti → B. microti
Line 86; a ddition → addition ?
Line 384; Please provide references. Is coinfection more common in canine babesiosis?
Line390; Babesia Canis → Babesia canis?Babesia vogeli?Babesia rossi?
Line 392-396; I understand that it can be detected as early as 6th day and as late as 8th day. What was the infection rate at that time? Could it not be detected by EM or PCR? As I mentioned in the introduction, it doesn’t convey much useful points compared to other methods. The same applies to the fact that it can be detected in the chronic infection on the 101th day. In addition, information on the dog’s condition at that time (general condition, red blood cell count, platelet count, etc.) is required.
Line 397-403; Is it possible to say that there are few false positives from this result? I don’t think the result of a match rate of 83% is high.
Author Response

(The authors gave the same response as above.)

Round 2
Reviewer 3 Report
Authors' responses for mentioned comments are satisfactory. Please pay extra attention to English grammar and spelling.
Reviewer 4 Report
Nothing